

# Age, growth, and recruitment patterns of juvenile ladyfish (*Elops* sp) from the east coast of Florida (USA)

Juan C. Levesque

Environmental Resources Management, Impact, Assessment, and Planning Division, Tampa, FL, United States

Corresponding author
Juan C. Levesque,
shortfin_mako_shark@yahoo.com

## ABSTRACT

Ladyfish (*Elops* sp) are a common and economically valuable coastal nearshore species found along coastal beaches, bays, and estuaries of the southeastern United States, and subtropical and tropical regions worldwide. Previously, ladyfish were a substantial bycatch in Florida's commercial fisheries, but changes in regulations significantly reduced commercial landings. Today, ladyfish are still taken in commercial fisheries in Florida, but many are also taken by recreational anglers. Life-history information and research interest in ladyfish is almost non-existent, especially information on age and growth. Thus, the overarching purpose of this study was to expand our understanding of ladyfish age and growth characteristics. The specific objectives were to describe, for the first time, age, growth, and recruitment patterns of juvenile ladyfish from the east coast of Florida (USA). In the Indian River Lagoon (IRL), annual monthly length-frequency distributions were confounded because a few small individuals recruited throughout the year; monthly length-frequency data generally demonstrated a cyclical pattern. The smallest were collected in September and the largest in May. Post-hoc analysis showed no significant difference in length between August and May, or among the other months. In Volusia County (VC), annual monthly length-frequency distribution demonstrated growth generally occurred from late-winter and spring to summer. The smallest ladyfish were collected in February and the largest in August. On average, the absolute growth rate in the IRL was 36.3 mm in 60 days or 0.605 mm day$^{-1}$. Cohort-specific daily growth rates, elevations, and coincidentals were similar among sampling years. Cohort-specific growth rates ranged from 1.807 in 1993 to 1.811 mm day$^{-1}$ in 1994. Overall, growth was best (i.e., goodness of fit) described by exponential regression. On average, the absolute growth rate in VC was 28 mm in 150 days or 0.1866 mm day$^{-1}$. Cohort-specific daily growth rates were significantly different among sampling years; however, the elevations and coincidentals were similar. Cohort-specific growth rates ranged from 1.741 in 1994 to 1.933 mm day$^{-1}$ in 1993. Mean ladyfish growth was best described by linear regression; however, natural growth was explained better by exponential regression. In the IRL, the corrected exponential growth equation yielded a size-at-age 1 of 156.0 mm SL, which corresponded to an estimated growth rate of 0.4356 mm day$^{-1}$. In VC, the corrected exponential growth equation yielded a size-at-age 1 of 80 mm SL corresponding to an estimated growth rate of 0.2361 mm day$^{-1}$.

## INTRODUCTION

Ladyfish (*Elops* sp) are a common nearshore species found along coastal beaches, bays, and estuaries of the southeastern United States (*Zale & Merrifield, 1989*; *McBride et al., 2001*), and subtropical and tropical regions worldwide (*Ugwumba, 1989*; *Brinda & Bragadeeswaran, 2005*). Seven *Elops* species have been identified worldwide (*Adams et al., 2013*); two  (*Elops saurus* and *Elops smithi*) are found in the western North Atlantic Ocean (*McBride & Horodysky, 2004*; *McBride et al., 2010*; *Adams et al., 2013*). Ladyfish have a specialized leptocephalus larval stage (*Gehringer, 1959*), which is uncommon to fish; most fish do not go through a metamorphosis stage after hatching (*Smith, 1989*). Approximately 800 species have a leptocephalus larval stage, but most are eels (*Greenwood et al., 1966*; *Smith, 1989*). Tarpon (*Megalops atlanticus*) and bonefish (*Albula vulpes*) are the only other economically and socially valuable fish that have a leptocephalus larval stage. Tarpon and bonefish support valuable recreational fisheries in the United States, Central America, and other subtropical/tropical regions worldwide (*Cooke & Philipp, 2004*; *Cooke et al., 2009*; *Fedler & Hayes, 2008*). Previously, ladyfish were a substantial portion of commercial landings in Florida, but changes in regulations during the mid-90s significantly reduced commercial landings of ladyfish (*Levesque, 2011*). Today, ladyfish are still taken in commercial fisheries in Florida, but many are also taken by recreational anglers (*Levesque, 2011*).

Understanding a species' life-history characteristics is necessary for making informed decisions and implementing successful management measures. Unfortunately, life-history information and research interest in ladyfish is almost non-existent, especially information on age and growth (*Adams et al., 2013*). Several brief notes (*Alikunhi & Rao, 1951*; *Gehringer, 1959*) and studies (*McBride et al., 2001*; *Levesque, 2014*) have reported information about age and growth, but knowledge is limited, speculative, and incomplete. Although *Levesque (2014)* described age and growth of juvenile ladyfish in Tampa (Florida), and *McBride et al. (2001)* reported the age and growth for larger size-classes in Tampa Bay and the Indian River Lagoon, these studies were somewhat restricted in terms of geography and analytical procedures. Currently, there are no studies that corroborate or validate age estimates of ladyfish. Given this management need, the overarching purpose of this study was to expand our understanding of ladyfish age and growth characteristics. The specific objectives were to describe, for the first time, age, growth, and recruitment patterns of juvenile ladyfish from the east coast of Florida (USA).

## MATERIAL AND METHODS

### Study area

Field-collections were made at numerous locations throughout the Indian River Lagoon (IRL) and Volusia County (VC (Tomoko River Basin, Ponce de Leon Inlet, and Mosquito Lagoon complex)). Field sampling was conducted by Florida Fish and Wildlife Conservation Commission (FWC), Fisheries Independent Monitoring (FIM), personnel at 21 (seines [8], trawls [11],  and gillnet [2]) pre-determined stations (i.e., fixed stations [FS]) in the IRL (Fig. 1) and 29 (seines [14] and trawl [15]) FS in VC (Fig. 2);  FS were

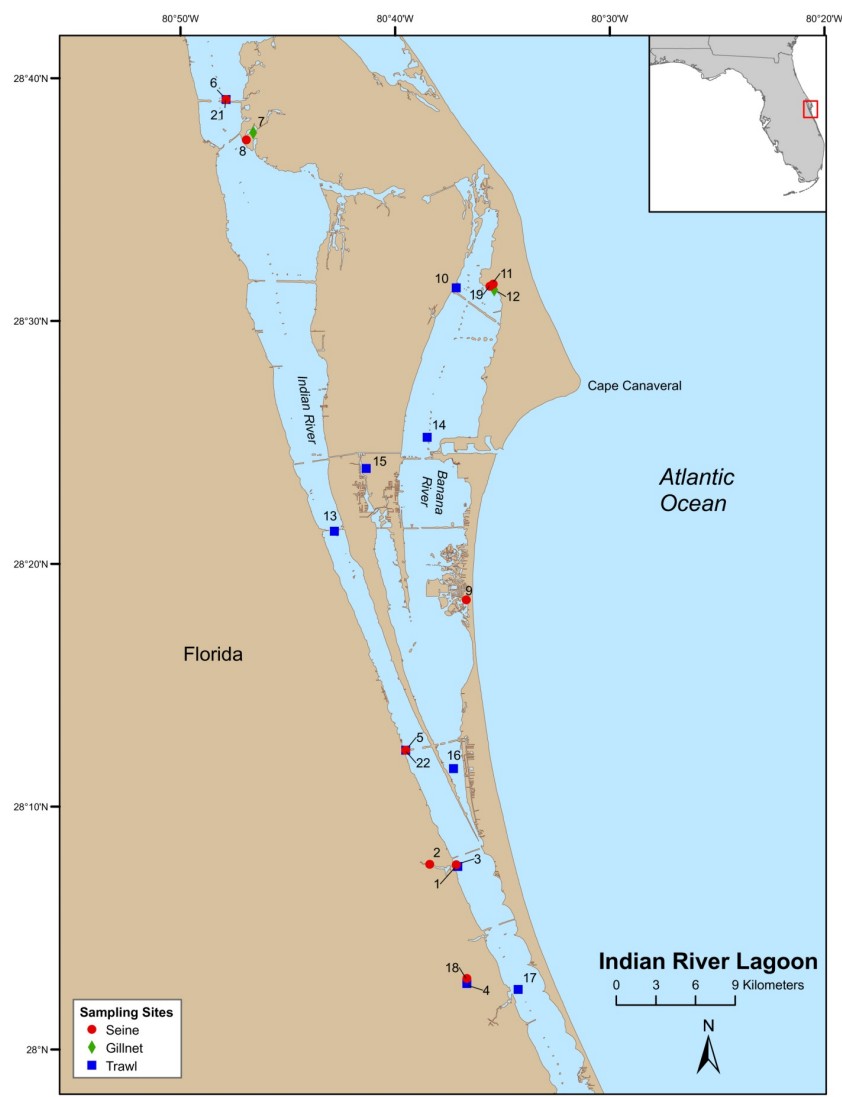

**Figure 1** Map of Indian River Lagoon sampling stations.

stratified by geographical location, habitat, and depth (*McMichael et al., 1995*). Further details on site descriptions are provided by *Levesque (2013)*.

## Gear and sampling methodology

Field sampling at FS was conducted once a month during daylight (i.e., the period between one hour after sunrise and one hour before sunset). Three haul repetitions were made at each station with a center-bag seine (21.3 m long by 1.8 m high; center bag constructed of 3.2 mm #35 knotless nylon Delta mesh). Based on the profile of the beach (i.e., bank slope) and water depth, one of three deployment methods (beach, boat, or offshore) were used to deploy the center-bag seine (i.e., seine) at each station (*McMichael et al., 1995*). The first deployment technique was the beach method. A beach deployment method was used when the water depth was shallow and the bank had either a gradual slope or no slope. The beach

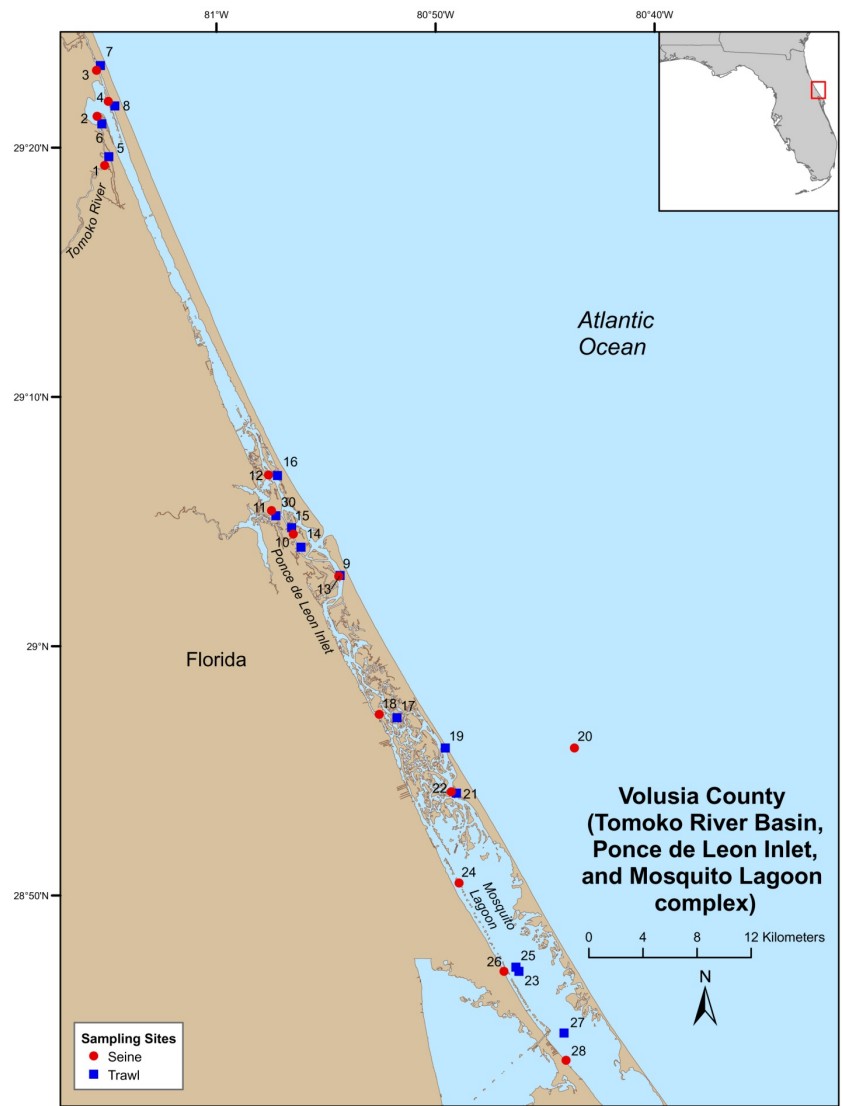

**Figure 2  Map of Volusia County sampling stations.**

deployment method consisted of the seine being pulled parallel to shore by two biologists for a total distance of 9.1 m; a 15.5 m line stretched between each seine pole was used to assure the net was being pulled the same inner-pole distance for every haul. The second deployment technique was the boat deployment method. A boat deployment method was used when the water was either to deep (water depth 0.7–1.2 m) or the bank was too steep to use a beach deployment. The boat deployment method consisted of deploying the seine from the stern in a semi-circular pattern along the bank. Once the seine was fully deployed, two biologists would pull the seine toward shore. The third and final deployment method was the offshore deployment method. An offshore deployment was used when there was either no available beach or it was too shallow to reach the beach bank by boat. The offshore deployment followed the same procedures as the beach deployment with one

minor difference; at the end of the 9.1 m distance, two biologists worked the seine using a stationary pivot pole to ensure the catch did not escape (*McMichael et al., 1995*). Given the seine dimensions and the distance traveled (9.1 m) along the beach, the total area sampled with the beach seine was 100 m$^2$.

## Data

The FWC used two experimental field sampling approaches in the 1990s to survey fish throughout Florida (*McMichael et al., 1995*): monthly FS and year-round stratified random sampling (SRS). The FWC conducted fisheries monitoring using a variety of sampling gears, such as center-bag seines, otter trawls, gillnets, blocknets, and dropnets. For these analyses, data was restricted to monthly FS collections of ladyfish collected with a center-bag seine because fewer juvenile ladyfish were collected with the SRS approach. Therefore, pooling the datasets (SRS and FS) could have bias the analyses by under- or over-estimating size-at-age. Also, most ladyfish collected by the SRS approach were larger and older than the pre-selected maximum cut-off length of 100 mm SL. Following *Levesque (2014)*, a maximum cut-off length of 100 mm SL was chosen because ladyfish larger than 100 mm SL could avoid some field sampling gear (i.e., small-mesh center-bag seines). After every net haul, fish were sorted, enumerated, and measured to the nearest 1 mm standard length (SL); a total of 20 individuals of every species were measured. It should be noted these data were collected prior to *McBride et al. (2010)* describing *Elops smithi*; *Elops saurus* and *Elops smithi* are both found on the east coast of Florida. Unfortunately, the FWC team was unaware at the time of the study that there were potentially two different species of ladyfish that could be found within the study area. As such, my findings only refer to the genus *Elops*.

## Statistical analysis

Data were evaluated for normality and homoscedacity (variance (equivalently standard deviation) are equal) using Kolmogorov–Smirnov (*Zar, 1999*) and Bartlett (*Bartlett, 1937a*; *Bartlett, 1937b*) tests, respectively. If the data passed the normality tests, then parametric procedures were followed; otherwise, the data were log-transformed [$\log(X + 1)$] to meet the underlying assumptions of normality (*Zar, 1999*). Non-parametric procedures were applied if the data could not meet the assumptions of normality after transformation. A *post-hoc* multiple comparison test was used to perform pairwise comparisons in the presence of significance at the 95% confidence level for either the Analysis of Variance (ANOVA) or Kruskal–Wallis non-parametric multi-sample tests. All analyses were conducted using Microsoft Excel® and Statgraphics Centurion XVI® Version 16.1. Statistical significance was defined as $p < 0.05$.

To estimate growth, monthly field collections of cohort lengths were categorized into 5 mm SL size classes, graphed, and evaluated. Descriptive statistics (e.g., mean, standard deviation, variance, standard error) were derived and cohorts identified using modal progression analysis (MPA); MPA consisted of plotting the mean SL and the collection date (*Petersen, 1892*). Before evaluating cohort modal progressions, a one-way ANOVA test was used to distinguish whether there was a significant difference in length among

months, years and locations. Annual ladyfish growth was estimated by regression analyses of the monthly geometric mean SL on capture date. Growth was described by linear (SL = slope [age] + $y$-intercept) and nonlinear regression. The coefficient of determination value was used to choose the most parsimonious (i.e., the model that best fit the data) growth model. Exponential growth regression was described with the following equation:

$$SL = L_o e^{Gt}$$

where, SL = standard length (mm); $G$ = instantaneous growth coefficient (per month); $L_o$ = initial SL (mm) size at first capture; $t$ = the time (per month) for the average individual in the length-class to achieve the indicated size.

The relative instantaneous growth coefficient ($G$) was estimated by calculating the average time individuals in a year-class attained a certain length (*Deegan, 1990*). The instantaneous growth coefficient was used to represent the average growth of the population during the time period (*Ricker, 1975*). The absolute daily growth rate was estimated by the following equation:

$$G = \Delta l(l_2 - l_1)/\Delta t(t_2 - t_1)$$

where, $l_2$ = SL (mm) at the end of a unit of time; $l_1$ = initial SL (mm) at time 0; $t_2$ = at the end of a unit of time (days); $t_1$ = initial time 0 (days).

Analysis of Covariance (ANCOVA) was used to determine if the slopes of the regression lines were significantly different (homogeneity of slopes assumption); significance criteria (homogeneity of y-intercepts and coincidental slopes and intercepts of the regression lines) was achieved when the parallelism of slopes assumption was met. If annual growth rates were equal, then the data were pooled. Following *Ricker (1975)*, it was assumed: (1) the population sampled had a normal distribution; (2) the size classes (captured) were not influenced by gear or sampling methods; (3) mortality was the only natural population influence; and (4) the population was resident to the sampling location (i.e., lack of immigration or emigration). In general, I assumed a steady-state and a closed-population. These basic assumptions are often applied to derive various life-history estimates, including mortality and recruitment (*Ziegler, Welsford & Constable, 2011*). Based on *Levesque (2014)* and *McBride et al. (2001)*, these population assumptions seemed reasonable because the data was limited to seine gear, and most of the sampling stations were located in ideal ladyfish habitat (*Eldred & Lyons, 1966*; *Gilmore et al., 1981*; *McBride et al., 2001*; *Florida Fish and Wildlife Conservation Commission, 2006*).

Growth and growth rates were evaluated to ensure estimates were realistic and biologically accurate given ladyfish have a metamorphic development (i.e., leptocephalus). Since ladyfish early development consists of the body shrinking before it transitions into the juvenile stage, estimating growth is somewhat challenging compared to most fish, especially if attempting to back-calculate size and age. If the derived size was unrealistic both in terms of recruitment and projected age-1 length, then size was corrected (y-intercept) to compensate for the unrealistic smaller predicted recruitment size and
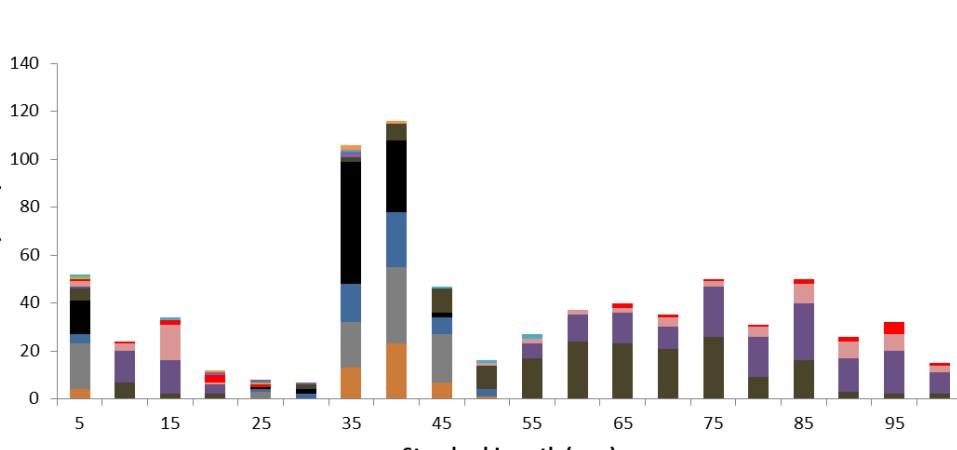

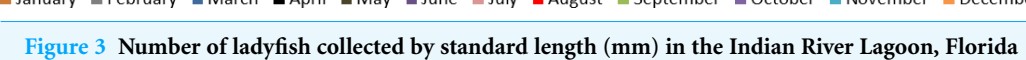

**Figure 3 Number of ladyfish collected by standard length (mm) in the Indian River Lagoon, Florida during 1991 through 1995.**

larger projected age-1 length. Using linear regression, the y-intercept of the exponential regression formula was corrected (standardized) to 21 mm SL to better reflect natural growth. The 21 mm SL was selected because it is generally the length ladyfish have before transitioning from the leptocephalus to the juvenile stage (*Alikunhi & Rao, 1951*; *Gehringer, 1959*). It is also the minimum size usually collected with a 3.2 mm #35 knotless Delta mesh beach seine. It should be noted that this mesh size seine can potentially capture smaller individuals, but 20 mm SL is a conservative size.

## RESULTS

### Length-frequency

A total of 767 juvenile ladyfish ranging from 1 to 99 mm SL ($\bar{x} = 48.8$ mm, S.D. $\pm 26.3$ mm) were collected in the IRL during 1991 through 1995. Annual monthly length-frequency distributions were confounded because a few small individuals were collected throughout the year; monthly length-frequency data generally demonstrated a cyclical pattern (Figs. 3 and 4). The smallest ladyfish ($\bar{x} = 12.5$ mm SL, S.D. $\pm 13.4$ mm, $n = 2$) were collected in September and the largest ($\bar{x} = 65.3$ mm SL, S.D. $\pm 28.2$ mm, $n = 174$) in May [$F(11, 753) = 31.87, P < 0.001$]. *Post-hoc* analysis showed no significant difference in length between August and May, or among the other months. Two separate one-way ANOVAs showed length during April [$F(2, 97) = 0.15, P = 0.86$] and June [$F(3, 50) = 2.35, P < 0.08$] was not significantly different among years; however, mean length in May was significantly different among sampling years [$F(2, 187) = 21.44, P < 0.001$]. The smallest ladyfish ($\bar{x} = 42.7$ mm SL, S.D. $\pm 16.73$ mm, $n = 44$) captured in May occurred in 1993 and the largest ($\bar{x} = 64.6$ mm SL, S.D. $\pm 14.01$ mm, $n = 98$) in 1995.
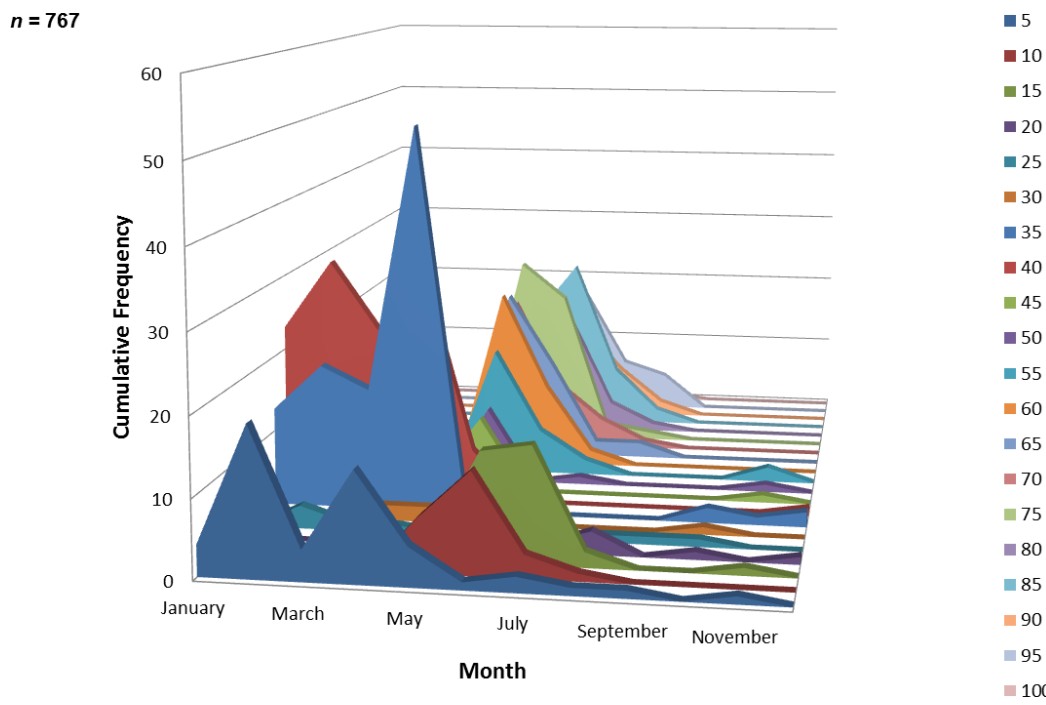

**Figure 4 Number and size of ladyfish collected by month in the Indian River Lagoon, Florida during 1991 through 1995.**

One hundred and sixty-nine juvenile ladyfish ranging from 2 to 99 mm SL ($\bar{x} = 34.3$ mm SL, S.D. $\pm 16.92$ mm) were collected in VC waters during 1993 through 1995. Annual monthly length-frequency distribution demonstrated that growth generally occurred from late-winter and spring to summer (Figs. 5 and 6). The smallest ladyfish ($\bar{x} = 19.3$ mm SL, S.D. $\pm 19.61$ mm, $n = 4$) were collected in February and the largest ($\bar{x} = 70.8$ mm SL, S.D. $\pm 34.24$ mm, $n = 4$) in August [$F(8, 160) = 6.04, P < 0.001$]. *Post-hoc* analysis showed no significant difference in length among September, October, March, January, April, May, June, and August. In addition, no significant difference in length was found among February, September, October, March, January, April, and May. Three separate one-way ANOVAs showed length in April [$F(2, 114) = 0.65, P = 0.52$], May [$F(2, 4) = 2.27, P = 0.22$], and June [$F(2, 10) = 1.88, P = 0.20$] was not significantly different among years.

## Length-frequency progressions

Ladyfish growth in the IRL was unable to be estimated by the progression of monthly cohort sizes as recruitment of smaller individuals occurred throughout the year. Therefore, for comparison purposes, and to eliminate recruitment bias (i.e., influx of small individuals), growth evaluations in the IRL were limited to catches occurring from April to June. This corresponded to the period when recruitment was not only consistent, but monthly mean size generally increased from one month to the next. The monthly instantaneous growth coefficient ranged from $-0.0677$ in 1995 to 0.094 in 1991. Absolute growth ranged from 0.55 in 1992 to 0.63 mm day$^{-1}$ in 1993 and 1994. On average, the absolute growth

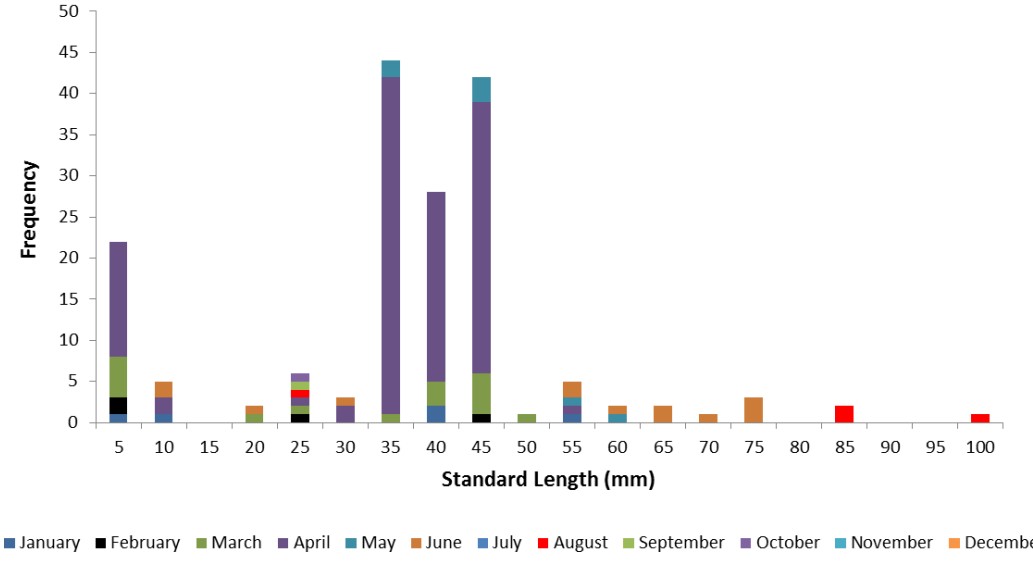

**Figure 5 Number of ladyfish collected by standard length (mm) in Volusia County, Florida during 1993 through 1995.**

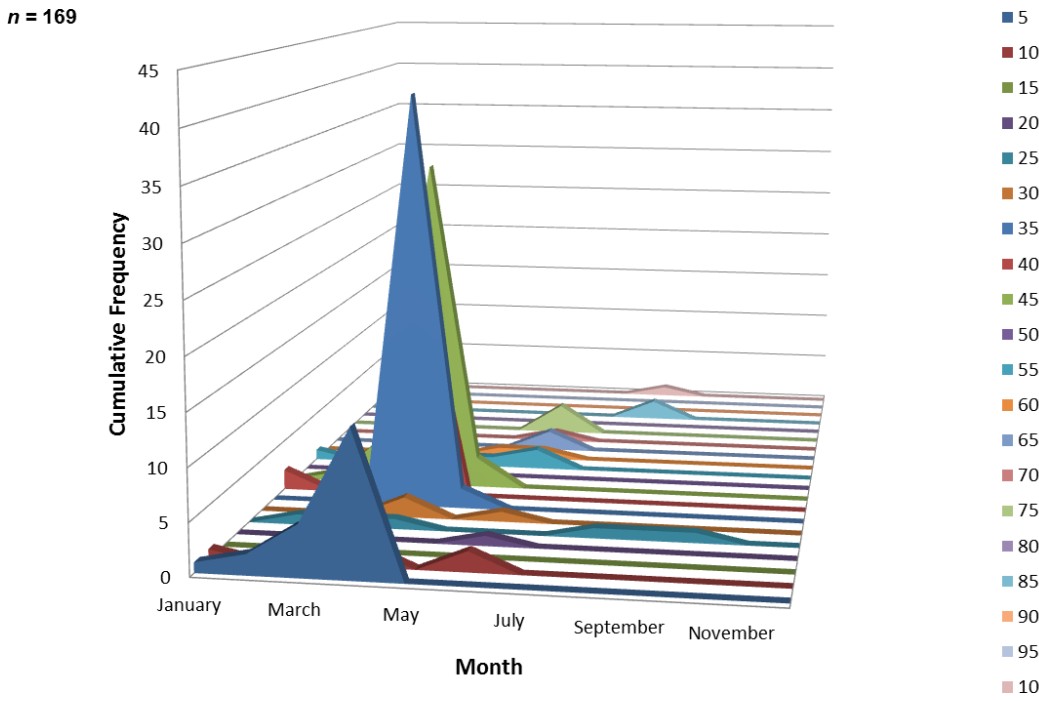

**Figure 6 Number and size of ladyfish collected by month in Volusia County, Florida during 1993 through 1995.**

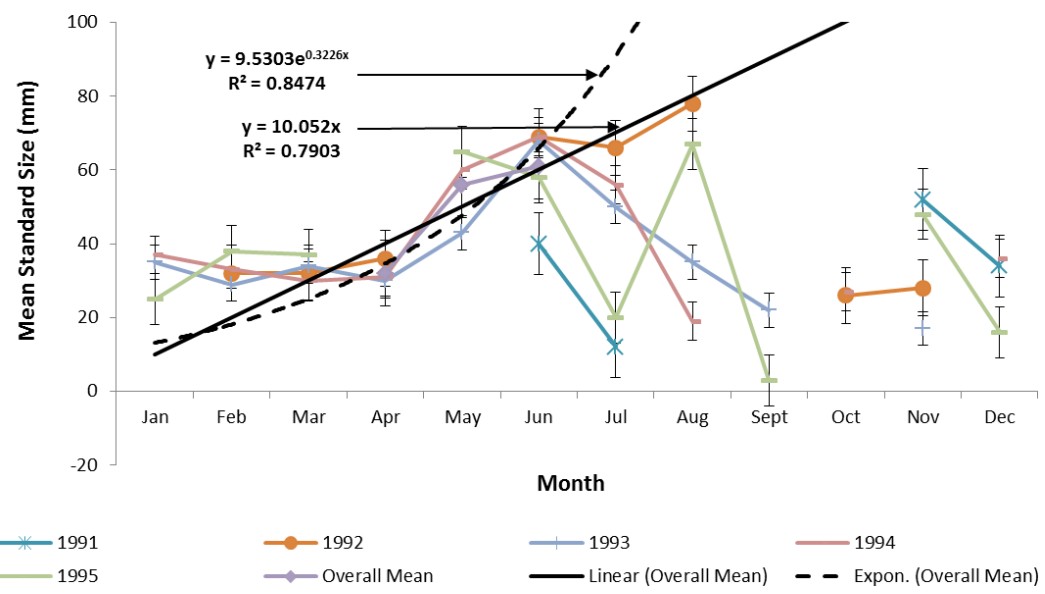

**Figure 7 Annual mean growth of juvenile ladyfish collected in the Indian River Lagoon, Florida during 1991 through 1995.**

rate was 36.3 mm in 60 days or 0.605 mm day$^{-1}$. Cohort-specific daily growth rates, elevations, and coincidentals (slopes and intercepts of the regression lines) were similar among sampling years $[F(1, 2) = 0.0035, P = 0.3146]$; $[F(1, 3) = 1.545, P = 0.2702]$; $[F(2, 2) = 0.5177, P = 0.3121]$, respectively. Cohort-specific growth rates ranged from 1.807 in 1993 to 1.811 mm day$^{-1}$ in 1994 ($\bar{x} = 1.811$ mm day$^{-1}$, S.D. $\pm 0.003$ mm day$^{-1}$). The overall growth was best (i.e., goodness of fit) described by an exponential regression having the formula: SL $= 9.5030^{0.3226 \, (\text{age})}$; $r^2 = 0.8474$ (Figs. 7 and 9). If the exponential trajectory rate was maintained over 365 days, ladyfish would attain a standard length of 457.5 mm corresponding to an estimated growth rate of 1.25 mm day$^{-1}$ (Tables 1 and 2). The corrected exponential growth equation yielded a size-at-age 1 of 156.0 mm SL, which corresponded to an estimated growth rate of 0.4356 mm day$^{-1}$ (Tables 1 and 2).

Estimating ladyfish growth from VC collections was also problematic because recruitment of small individuals occurred throughout the year and the estimated growth rate varied among sampling years. Therefore, to compensate for the recruitment of small individuals in VC, growth evaluations were limited to catches occurring from March to August. The monthly instantaneous growth coefficient ranged from $-0.3061$ in 1995 to 0.3324 in 1994. Absolute growth ranged from 0.3833 in 1993 to 0.5833 mm day$^{-1}$ in 1994. On average, the absolute growth rate was 28 mm in 150 days or 0.1866 mm day$^{-1}$. Cohort-specific daily growth rates were significantly different among sampling years $[F(2, 15) = 3.6921, P = 0.0497]$; however, the elevations and coincidentals were similar $[F(2, 17) = 0.4349, P = 0.3927]$; $[F(4, 15) = 2.1324, P = 0.1402]$, respectively. Cohort-specific growth rates ranged from 1.741 in 1994 to 1.933 mm day$^{-1}$ in 1993 ($\bar{x} = 1.837$ mm day$^{-1}$, S.D. $\pm 0.14$). Mean ladyfish growth was best (i.e., goodness of fit) described by a linear regression having the formula: SL $= 5.4429$ (age [days]) $+ 11.1$;

Table 1 **Corrected and non-corrected juvenile ladyfish growth rates and size-at-age 1 (without compensating for time required for leptocephalus to metamorphosis from egg to juvenile) based on length-frequency analysis by location.** The annual mean growth rate and size-at-age 1 was estimated by pooling the data for each location. The y-intercept of the exponential regression formula was corrected to 21 mm SL to better reflect natural growth (shaded cells). Locations are as follows: Indian River Lagoon (IRL), Tampa Bay (TB), Volusia County (VC (Tomoko River Basin, Ponce de Leon Inlet, and Mosquito Lagoon complex)), and Little Manatee River (LMR). Data for TB and LMR was reported by *Levesque (2014)*.

| Year | Growth rate (mm/day) | | | | Size-at-age 1 (mm SL) | | | |
|---|---|---|---|---|---|---|---|---|
| | TB | IRL | VC | LMR | TB | IRL | VC | LMR |
| 1988 | – | – | – | 0.0001 | – | – | – | 0.0449 |
| | | | | 0.0259 | | | | 9.5 |
| 1989 | 1.76 | – | – | 3.976 | 643.2 | – | – | 1451.5 |
| | 0.9175 | | | 0.5658 | 334.9 | | | 206.5 |
| 1990 | 1.41 | – | – | 0.0382 | 515.2 | – | – | 13.9 |
| | 0.5671 | | | 0.1284 | 207.0 | | | 46.9 |
| 1991 | 0.98 | 0.1102 | – | – | 358.2 | 40.2 | – | – |
| | 0.4123 | 0.0986 | | | 150.5 | 35.9 | | |
| 1992 | 0.58 | 0.1173 | – | – | 211.5 | 42.8 | – | – |
| | 0.4452 | 0.1172 | | | 162.5 | 42.8 | | |
| 1993 | 11.78 | 0.0711 | 0.0980 | – | 4301.3 | 25.9 | 35.8 | – |
| | 0.4378 | 0.1132 | 0.1224 | | 159.8 | 41.3 | 44.7 | |
| 1994 | 2.74 | 0.1074 | 0.9934 | – | 1001.1 | 39.2 | 362.8 | – |
| | 0.6693 | 0.1339 | 0.1568 | | 244.3 | 48.8 | 57.3 | |
| 1995 | 1.20 | 0.0542 | 0.0630 | – | 436.8 | 19.8 | 23.0 | – |
| | 0.5304 | 0.0754 | 0.1224 | | 193.6 | 27.5 | 44.7 | |

Table 2 **Juvenile ladyfish growth rates and size-at-age 1 based on length-frequency analysis in Florida waters by location.** For comparison purposes, the direct method growth rate determined by captive rearing (*Levesque, 2014*) is shown along with available ladyfish age and growth estimates from previous studies (*Alikunhi & Rao, 1951*; *Gehringer, 1959*; *McBride et al., 2001*). The overall mean growth rate and size-at-age 1 was estimated by pooling the data for each location. The y-intercept of the exponential regression formula was corrected to 21 mm SL to better reflect natural growth (shaded cells). Locations are as follows: Indian River Lagoon (IRL), Tampa Bay (TB), Volusia County (VC (Tomoko River Basin, Ponce de Leon Inlet, and Mosquito Lagoon complex)), and Little Manatee River (LMR). Data for TB and LMR was reported by *Levesque (2014)*.

| Age determination method | Growth rate (mm/day) | | | | Size-at-age 1 (mm SL) | | | |
|---|---|---|---|---|---|---|---|---|
| | TB | IRL | VC | LMR | TB | IRL | VC | LMR |
| *Present study:* length-frequency analysis (data pooled) | 1.11 | 1.25 | 0.2947 | 1.04 | 403.6 | 457.5 | 107.6 | 380.9 |
| | 0.9101 | 0.4356 | 0.2356 | 0.3882 | 332.2 | 156.0 | 80.0 | 141.7 |
| *Levesque (2014)* | | | 0.8134 | | | | 296.9 | |
| *Alikunhi & Rao (1951)* | | | 0.78 | | | | 284.7 | |
| *Gehringer (1959)* | | | 0.63 | | | | 229.9 | |
| *McBride et al. (2001)* | | | 0.5479–0.8219 | | | | 200–300 | |

$r^2 = 0.8711$. However, natural growth was explained better by the exponential regression formula: $SL = 16.846^{0.1545 \, (age)}$; $r^2 = 0.8659$ (Figs. 8 and 9). If the exponential trajectory rate was maintained over 365 days, ladyfish would attain a standard length of 107.6 mm corresponding to an estimated growth rate of 0.2951 mm day$^{-1}$ (Tables 1 and 2). The corrected exponential growth equation yielded a size-at-age 1 of 80 mm SL corresponding to an estimated growth rate of 0.2361 mm day$^{-1}$ (Tables 1 and 2).

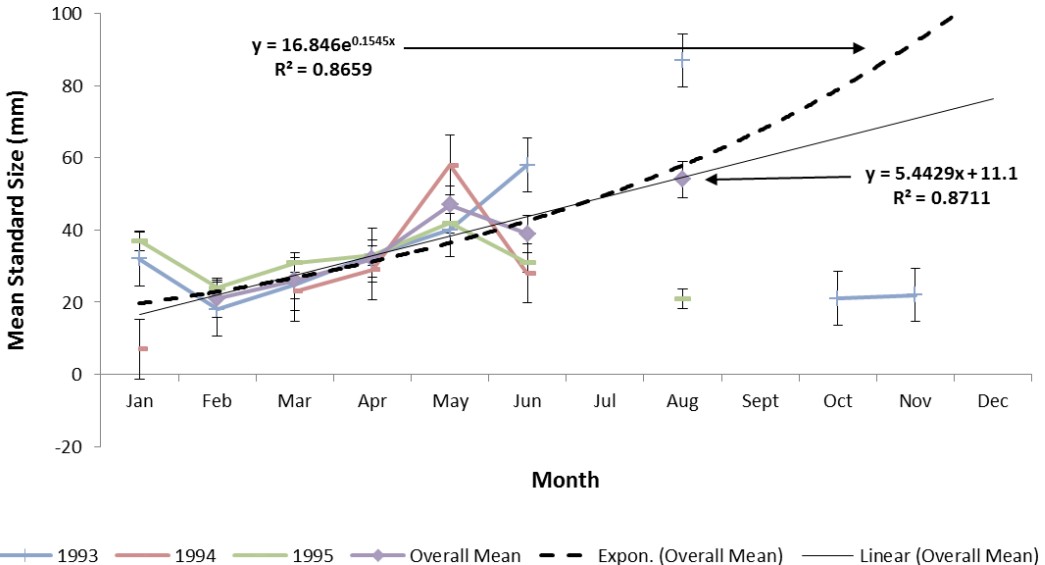

**Figure 8** **Annual mean growth of juvenile ladyfish collected in Volusia County, Florida during 1993 through 1995.**

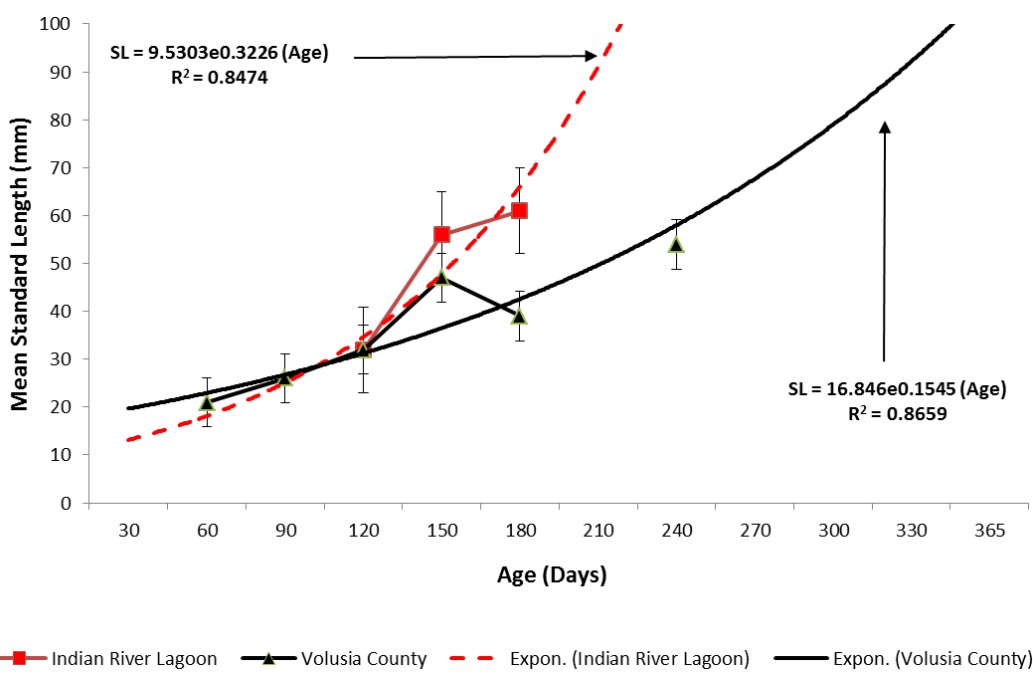

**Figure 9** **Overall mean growth of juvenile ladyfish collected in the Indian River Lagoon and Volusia County, Florida during 1991 through 1995.**

## DISCUSSION

*Laslett, Eveson & Polacheck (2004)* indicated the progression of cohort growth can be modeled under certain circumstances, but using length-frequency data to estimate fish growth is not always a straightforward approach. Realistic age and growth estimates for

juvenile ladyfish using length-frequency data were derived, but I did have to consider monthly and annual recruitment patterns in my analyses. Therefore, the interpretation and discussion of these results are reported with some reservation since the length-frequency data were rather unpredictable and ages were not directly validated with hard body parts (i.e., otoliths). Findings showed that the recruitment phase was inconsistent and prolonged from year-to-year in the IRL and VC waters, which made predicting growth more difficult since data could not be pooled. Overall, monthly recruitment varied somewhat due to the immigration of a few individuals. It is difficult to explain whether these individuals were either *Elops saurus* or *Elops smithi* since both species are found on the east coast of Florida. Also, understanding the results was challenging because these data were collected prior to *McBride et al. (2010)* described the new species; unfortunately, field staff were unaware of the two *Elops* species potentially occurring at the same time in the study area. Available information suggest that *E. smithi* have an extended recruitment period and it could be year round (*McBride & Horodysky, 2004*; *McBride et al., 2010*), which might explain the inconsistent pattern in recruitment. *Laslett, Eveson & Polacheck (2004)* also stated that variability in annual growth needs to be considered during length-frequency analyses since environmental conditions might be more favorable for growth in some years, especially between and among species. Although information describing growth for *E. smithi* is unavailable, it is possible that environmental conditions (i.e., temperature) may influence the two *Elops* species differently. Interestingly, the data showed that mean ladyfish size, during some months of the recruitment phase, varied among sampling years in the IRL, but not in VC. Nonetheless, regression analysis showed there was no significant difference in ladyfish growth among sampling years in the IRL.

*McBride et al. (2001)* reported ladyfish in the IRL attain between 250 and 270 mm SL by age-1. However, my findings showed that the growth rate and projected age-1 length was significantly smaller (156.0 mm SL [IRL] and 107.6 mm SL [VC]) than their estimates. Strangely, I derived different age-1 estimates for the IRL and VC despite the close proximity between the two areas. It is highly probable that the difference in age-1 length was related to differences in recruitment of *E. saurus* and *E. smithi*. However, another possible explanation for the difference in predicted age-1 length could have been attributed to either differences in environmental factors (e.g., water temperature, pH, salinity, and dissolved oxygen) or habitat between the two areas. It is probably unlikely that environmental factors were significantly different between the areas given their geographic similarity and close proximity, so it is possible that there were slight differences in suitable habitat or prey availability; fish grow differently depending on the habitat (*Sogard, 1992*; *Phelan et al., 2000*). Again, as stated above, these data were collected before *E. smithi* was described by researchers so there is no way to thoroughly explain why I found differences in predicted age-1 size between the IRL and VC.

Regardless of the reasons why the data displayed some variability in annual monthly size, growth was reasonably modeled using length-frequency data; this confirms the applicability of length-frequency data for estimating annual growth. Though the projected age-1 length for VC (108 mm SL) was possibly misleading given the small sample size,

the overall projected age-1 length (108–458 mm SL) seemed reasonable. In TB (*Levesque, 2014*) and the IRL (this present study), the projected lengths at age-1 were 404 and 458 mm SL, respectively. However, when growth rates were corrected (y-intercept) to compensate for the unrealistic smaller predicted recruitment size and larger projected age-1 length, age-1 length were 332 mm SL for TB and 159 mm SL for the IRL. I would like to point out that the predicted age-1 length in TB was 52% longer than the length predicted for the IRL, so it is probable that corrected (y-intercept) length (21 mm) was overestimated. If the corrected size was adjusted to a lower value (15.5 mm SL), then the projected age-1 length would be 239 mm SL, which is still a smaller (28%) age-1 size than predicted by *Levesque (2014)* for TB. Thus, it appears ladyfish from the east coast of Florida are either smaller at age-1 than on the west coast (i.e., TB) or the projected growth regression formula was inaccurate or misleading. Based on field-collections, it is more probable that the corrected growth rate was accurate since the projected recruitment size (y-intercept) value of 15.5 mm SL was within the size range of individuals collected during the peak recruitment phase. It is difficult to speculate why there was a difference in predicted age-1 size between the two east coast areas, but it is likely that it was related to sample size or the presence of two *Elops* species. This present study derived a different estimated ladyfish age-1 size than *McBride et al. (2001)*, which emphasizes how differences in data treatment can affect the outcome. For instance, this study evaluated ladyfish collected with a center-bag seine since the objective was to evaluate juvenile ladyfish sizes (<100 mm SL) rather than all life-stages (*McBride et al., 2001*).

My length-frequency derived age-1 size estimates were similar to those reported by *Levesque (2014)* for captive reared ladyfish. Overall, length-frequency proved to be a satisfactory approach for estimating juvenile ladyfish age and growth from east coast waters of Florida. Few researchers have reported age and growth estimates for ladyfish, so it is difficult to compare these findings to others, but it appears that ladyfish (*Elops* sp) in the western North Atlantic Ocean (*McBride et al., 2001*; *Levesque, 2014*) grow faster than ladyfish (*E. affinis* and *E. lacerta*) in other regions (*Blake & Blake, 1981*; *Ugwumba, 1989*).

## CONCLUSIONS

Understanding a species' life-history characteristics is necessary for making informed decisions and implementing successful management measures. My findings offer insight into juvenile ladyfish growth, and demonstrate the usefulness of the Petersen method for estimating age and growth. These findings show that growth can be reasonably modeled through indirect methods (i.e., length-frequency progression), but results should be viewed with caution, particularly if there is variability in mean length during the recruitment period (within and among locations). Although it's not recommend that the Petersen approach be applied to species with an extended recruitment period, it is possible to derive reasonable growth estimates as long as appropriate analytical steps are followed. As evident in this study and others (*McBride et al., 2001*; *Levesque, 2014*), derived growth rates were sensitive to analyses, so it is recommended that researchers use long-term datasets when attempting to estimate growth from alternative methods.

In general, researchers should consider evaluating at least a 2–4 year time-series to resolve inter-annual trends, but the time-series length depends on various biological and environmental factors (e.g., local variability, geographical location, sampling gear, habitat, species, size-class, and the number of replicates).

## ACKNOWLEDGEMENTS

Special thanks are owed to the Fisheries Independent Monitoring (FIM) staff of the Florida Fish and Wildlife Research Institute; thank you for your dedicated field sampling, sorting, and gear maintenance efforts. I especially thank B McMichael and T McDonald for kindly providing access to the FIM data. Also, I thank C DeCurtis and B Reiser for providing editorial comments and edits that greatly improved the quality of this manuscript. I also thank K Knight and P Gehring for providing GIS graphics support. Lastly, I thank C Hager and an anonymous reviewer for their editorial and technical critique of the article.

### Funding

This work was supported in part by funding from Florida saltwater fishing license sales and the Department of Interior, US Fish and Wildlife Service, Federal Aid for Sportfish Restoration Project Number F-43 to the Florida Fish and Wildlife Conservation Commission. Sampling in the Little Manatee River was made available through grants CM-254 and CM-280 from the Department of Environmental Regulation, Office of Coastal Management, with funds made available through the National Oceanic and Atmospheric Administration; Florida Department of Environmental Regulations, Office of Coastal Management under the Coastal Zone Management Act of 1972, as amended. The funders had no role in study design, data collection and analysis, decision to publish, or preparation of the manuscript.

### Grant Disclosures

The following grant information was disclosed by the author:
Florida saltwater fishing license sales.
Department of Interior, US Fish andWildlife Service.
Federal Aid for Sportfish Restoration Project.
Florida Fish and Wildlife Conservation Commission: F-43.
National Oceanic and Atmospheric Administration; Florida Department of Environmental Regulations, Office of Coastal Management: CM-254, CM-280.

### Competing Interests

The author declares there are no competing interests.

### Author Contributions

- Juan C. Levesque conceived and designed the experiments, performed the experiments, analyzed the data, contributed reagents/materials/analysis tools, wrote the paper, prepared figures and/or tables, reviewed drafts of the paper.

## Supplemental Information

Supplemental information for this article can be found online at http://dx.doi.org/10.7717/peerj.1392#supplemental-information.

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
