# Peer review of "Age, growth, and recruitment patterns of juvenile ladyfish (Elops sp) from the east coast of Florida (USA)"

_PeerJ, doi:10.7717/peerj.1392_

## Round 0.1 · original submission · Minor Revisions

This is a nice manuscript which will benefit from incorporating reviewers comments, both on content and format. Please make sure to address all the comments from the reviewers, in particular please include all clarifications as requested, discuss temperature as a potential factor, discus the weaknesses of your experimental design, support your statements by references and revise the figure to make it clearer,

Reviewer 1 ·

Basic reporting

Line 81. The FWC and FIM acronyms are not defined at their first use. FWC is defined later on line 112. FIM is not explicitly defined until the acknowledgements, although “fisheries independent monitoring” is mentioned on line 114. Can these acronyms be defined when they are first used?

Line 111. This is the only subsection that has been numbered.

Line 123. The last part of this sentence is a bit ambiguous. The problem is the “20 individuals” in parenthesis: does this mean that only 20 individuals were measured after each haul, or that a maximum of 20 individuals per length class were included in the data, or something else? Can you please make this sentence more clear? I'm also curious about the number (average and range) of fish present in the hauls because the sampling areas seem quite small, perhaps an indication of this could be included?

Line 183. As written, this sentence could mean that 21 mm is the typical length either just before or just after metamorphosis. I think that 21 mm is the usual length of ladyfish after metamorphosis into the juvenile stage, but could you alter the sentence to make this more clear? Something as simple as changing “have transitioned” to “have upon transitioning”.

Line 308. “...can be modelled...”

Line 316. Change “do” to “due”.

Line 317. I think that the possibility of this analysis having been applied to two separate species is important. Although you mention that two species are found in the north-western Atlantic Ocean in the introduction, it is not clear that these two species both occur within the study region (Florida) until the discussion, and I was surprised when I got to line 317. I think it would be useful to know that the data could have contained two separate ladyfish species before reading the results section, so could this be mentioned during the introduction or materials and methods sections?

Line 323. I suppose that environmental conditions may also have varying affects on the growth of the two different species. This may be worth a mention in the discussion.

Line 340. “... projected growth rates at ...” needs changed to “... projected lengths at ...”

The colour scheme for the bar plots (figs. 3 & 5) makes the graphs quite difficult to follow. The colours used in several months are very similar, with the months spread throughout the year (e.g. February, March, July and November in fig. 3), so it is hard to separate the months. The months that are mentioned in the text as the months with the smallest and largest sampled fish (May/September fig 3. and February/August fig. 5) have been given the same colours in these graphs, and no other months share these colours, so they are relatively easy to distinguish. Representing each month with a more distinct colour, or using a smooth colour scale, could make these plots more accessible. This was not a problem in figs. 4 & 6 because the colours indicated length classes, and the lengths were shown in order on the graphs so could be counted if necessary.

Experimental design

No comment

Validity of the findings

No comment

Additional comments

The research presented in this paper is interesting and useful. The analysis appears to be sound, and I think that this paper is publishable. I have only a few comments (mostly about typos and unclear sentences), and these could be dealt with quickly and easily.

·

Basic reporting

Submission policies are a PeerJ issue not a scientific reviewers but it appears that the paper is presented in a reasonable template and that data sharing has occurred.

In this case, the raw data has been so transformed however that unless the reviewer wanted to reanalyze the data it is not that useful.

Figures are fine though a little busy.

Experimental design

The investigator has done the best he could with the field data. The central weakness of the approach hinges on the fact that small individuals recruited into the study sites and length frequency data demonstrated a cyclical pattern. Ricker's (1975) assumption of a resident population with no immigration or emigration is thus violated.

The author goes on to address this issue by sub-setting the data set so that this assumption is met and I believe this was done in a reasonable manner but not without costs.

Validity of the findings

To eliminate recruitment bias due to cyclical recruitment (which is one way to minimize your risk of violating one of the central assumptions) IRL data was reduced to catches from April to June (leaving only 3 points for slope calculation) and VC from March to August. Growth was then calculated and compared between sites based on slope. In theory duration of the study period between sites is less important due to this approach however fish are cold blooded and their growth is dependent upon temperature. The author acknowledges this in the discussion but not to my satisfaction with regard to how this factor relates to this study.

How do these systems compare in temperature? If the two systems are thermally equivalent the approach holds and differences in growth may be realistically due to habitat variation. However, if temperature varies due to your temporal sub-setting of data (April to June vs March to August) or natural system characteristics, growth may be thermally driven and independent of habitat and thus there are no differences between your sites?

These are not fatal flaws but temperature needs to be considered and discussed with respect to these systems and your results as appropriate. We know fish grow differently in varied habitats the important point is why?

Additional comments

I would like to see more citations supporting the population assumptions (line 173-174) and the assertion that sample locations were in ideal ladyfish habitat. Citing oneself to support claims central to analytical assumptions is not sufficient unless no other similar work has been done (work does not have to be the same species). This is especially true when you have to find statistical methods of manipulating data to meet these assumptions.

Citation needed on line 184 referencing transition to juvenile stage.
Line 201 verb like was should follow May.
Line 308 growth can be (no be )

---

## Round 0.2 · accepted · Accept

All reviewers comments have been sufficiently addressed. This is an interesting paper contributing to our knowledge of Elops sp.